

# The role of fMRI in the mind decoding process in adults: a systematic review

Sahal Alotaibi[1], Maher Mohammed Alotaibi[1], Faisal Saleh Alghamdi[1], Mishaal Abdullah Alshehri[1], Khaled Majed Bamusa[1], Ziyad Faiz Almalki[1], Sultan Alamri[1], Ahmad Joman Alghamdi[1], Mohammed Alhazmi[2], Hamid Osman[1] and Mayeen U. Khandaker[3,4,5]

[1] Department of Radiological Sciences, College of Applied Medical Sciences, Taif University, Taif, Saudi Arabia
[2] Public Security Administration of Medical Services, Ministry of Interior, Riyadh, Saudi Arabia
[3] Faculty of Graduate Studies, Daffodil International University, Dhaka, Dhaka, Bangladesh
[4] Applied Physics and Radiation Technologies Group, CCDCU, School of Engineering and Technology, Sunway University, Bandar Sunway, Malaysia
[5] Department of Physics, College of Science, Korea University, Seoul, Republic of South Korea

## ABSTRACT

**Background**. Functional magnetic resonance imaging (fMRI) has revolutionized our understanding of brain activity by non-invasively detecting changes in blood oxygen levels. This review explores how fMRI is used to study mind-reading processes in adults.
**Methodology**. A systematic search was conducted across Web of Science, PubMed, and Google Scholar. Studies were selected based on strict inclusion and exclusion criteria: peer-reviewed; published between 2000 and 2024 (in English); focused on adults; investigated mind-reading (mental state decoding, brain-computer interfaces) or related processes; and employed various mind-reading techniques (pattern classification, multivariate analysis, decoding algorithms).
**Results**. This review highlights the critical role of fMRI in uncovering the neural mechanisms of mind-reading. Key brain regions involved include the superior temporal sulcus (STS), medial prefrontal cortex (mPFC), and temporoparietal junction (TPJ), all crucial for mentalizing (understanding others' mental states).
**Conclusions**. This review emphasizes the importance of fMRI in advancing our knowledge of how the brain interprets and processes mental states. It offers valuable insights into the current state of mind-reading research in adults and paves the way for future exploration in this field.

# INTRODUCTION

Functional magnetic resonance imaging (fMRI) is a non-invasive neuroimaging technique that measures brain activity by identifying changes in blood oxygenation levels. It relies on the principle that neural activity is accompanied by alterations in blood flow and oxygenation in specific brain regions. During a brain scan, the individual lies inside a magnetic resonance imaging (MRI) scanner, which uses a strong magnetic field and radiofrequency pulses to generate detailed images of the brain's structure. The fMRI

Corresponding author
Mayeen U. Khandaker,
mayeenk@sunway.edu.my

component of the scan focuses on capturing functional information by measuring blood oxygenation level-dependent (BOLD) signals (*Heeger & Ress, 2002*).

The fMRI has revolutionized our understanding of the human brain by allowing researchers to peer into its intricate workings in real-time. Among its many applications, one of the most captivating is its potential to decode the inner workings of the mind (*Alotaibi et al., 2023*). In this context, "fMRI mind-reading/decoding" refers to the use of functional magnetic resonance imaging to explore two interconnected aspects of cognitive neuroscience: (I) directly decoding a person's cognitive states, intentions, or mental imagery from neural activity patterns, and (II) investigating how individuals interpret or infer the mental states of others through observed behavior or neural activity. These two areas address different aspects of mind-decoding: the former focuses on directly extracting mental content from brain activity, while the latter delves into understanding the brain processes involved in mental state attribution. Clarifying these distinctions is crucial, as both approaches contribute to our understanding of how fMRI can illuminate the workings of the human mind. This fascinating capability has led to speculation about the ability of fMRI to "read minds", offering insights into thoughts, intentions, and even emotions (*De Charms, 2008*). While the concept of fMRI mind-decoding may evoke images of science fiction, its actual capabilities and limitations provide a rich ground for exploration and discussion.

At the heart of fMRI lies the principle of detecting changes in blood flow and oxygenation levels in response to neural activity. By measuring these changes, researchers can pinpoint regions of the brain associated with specific cognitive processes or mental states (*De Charms, 2008*). Through sophisticated analysis techniques, such as multivariate pattern analysis (MVPA) and machine learning algorithms, fMRI data can be decoded to infer the content of a person's thoughts or intentions, albeit with varying degrees of accuracy (*Norman et al., 2006*).

The implications of fMRI mind-decoding extend far beyond the realm of neuroscience. In fields like psychology, psychiatry, and neurology, it offers potential insights into mental health disorders, decision-making processes, and even lie detection (*De Charms, 2008*). Moreover, the ability to decode neural activity raises profound ethical questions regarding privacy, autonomy, and the boundaries of individual consciousness.

While fMRI mind-decoding holds promise, it also faces significant challenges and limitations. The complexity of neural activity, the variability between individuals, and the ethical considerations surrounding privacy and consent all pose hurdles to its practical implementation. Nonetheless, the ongoing research in this field continues to push the boundaries of our understanding of the human mind and raises stimulating inquiries about the nature of consciousness and the future of brain-machine interfaces (*Lee et al., 2010*).

While substantial progress has been made in fMRI-based mind-decoding, certain psychological functions within healthy populations, such as empathy, theory of mind, and emotional regulation, remain underexplored in terms of how they are represented and processed in the brain. Specifically, the neural mechanisms underlying these complex social cognition functions require deeper investigation to advance our understanding of

mind-decoding capabilities. This systematic review addresses these gaps by examining how fMRI can shed light on the neural foundations of mind-decoding in healthy adults, focusing on cognitive processes that are essential for social interaction and emotional processing. This paper aims to understand the role of fMRI in mind decoding process in adults. Understanding the connection between fMRI and mind decoding is crucial as technology is constantly developing, and with the emergence of artificial intelligence, the world has become constantly talking about artificial intelligence.

This research investigates the potential of fMRI as a crucial tool for understanding adult minds, particularly in the realm of "mind-decoding". We hypothesize that fMRI offers a significant advantage over past techniques due to its non-invasive nature and ability to capture brain activity in real-time with greater safety and comfort for participants. Our hypothesis is based on a comprehensive review of prior studies exploring the impact of fMRI on mind decoding/mind-reading research in adults. These studies demonstrate the versatility of fMRI, as researchers have employed various mind-reading techniques across different adult populations. Previously, mind decoding techniques often relied on invasive procedures like implanting electrodes in the brain. These methods, while offering some insights, posed significant limitations and ethical concerns.

The field of neuroscience is experiencing a surge in research focused on unlocking the key to mind-decoding. Notably, a search of the current research in thought detection with fMRI has uncovered numerous recent discoveries in neuroscience that are bringing researchers closer to deciphering the key to mind-decoding. Each of these studies utilizes fMRI to observe brain activity in real-time. Many different experiments and methods related to our topic were conducted on adults, regardless of a direct or indirect relationship to our topic. However, it's crucial to distinguish between directly relevant and tangential research in this area. While some studies (*Rainey, 2024*; *Villarroya, 2013*; *Rose, 2016*; *Hsu et al., 2015*; *Beaucousin et al., 2007*) showcase a diverse range of research within neuroscience, others explore areas outside the core focus of adult mind-decoding using fMRI. For instance:

- "Rights and wrongs in talk of mind reading technology" (*Rainey, 2024*): This study discusses the applying of large language models in mind-reading technology, which is relevant but falls outside the scope of directly investigating fMRI's role.
- "The challenges of neural mind-reading paradigms" (*Villarroya, 2013*): This study explores the theoretical and methodological difficulties associated with mind-reading research, which is valuable but doesn't directly address fMRI's application.
- "Reading the human brain: how the mind became legible" (*Rose, 2016*): This study offers a broader perspective on brain imaging techniques and their contribution to understanding the mind, which provides valuable context but doesn't directly focus on fMRI and mind-decoding in adults.
- "The emotion potential of words and passages in reading harry potter - An fMRI study" (*Hsu et al., 2015*) and an FMRI study of emotional speech comprehension (*Beaucousin et al., 2007*): These references explore the neural basis of emotional processing, which while related to social cognition, deviate from the core focus of adult mind-decoding.

To gain deeper insights into the potential of fMRI for mind-decoding in adults, future research should focus on studies that directly investigate the decoding of thoughts and intentions using fMRI in adult populations. This focused approach will allow us to critically evaluate the strengths and limitations of fMRI in this specific domain.

## METHODS

In this section of the systematic review, we initiate by addressing a significant question that may arise in the reader's mind: Why is a systematic review beneficial for our topic? Systematic reviews play a pivotal role in enhancing our comprehension of fMRI studies pertaining to mind decoding through several key aspects. Firstly, they synthesize available evidence, providing readers with a comprehensive overview of the current state of knowledge in the field. Secondly, they identify trends, encompassing variations in methodologies, experimental paradigms, and the neural correlates that have been identified. Lastly, systematic reviews serve to validate our hypothesis, which speculates the irreplaceable role of fMRI in the process of mind decoding.

### Research question and hypothesis

This systematic review employs the PICO framework to formulate a research question regarding the role of fMRI in understanding adult mind-decoding (*Schardt et al., 2007*). It is defined as follows: Population (P): Adults in general. Interventions/Exposure (I): Utilization of fMRI techniques during mind decoding tasks. Comparison (C): No specific comparison is required for this research question, as the focus is on understanding the role of fMRI in the mind-decoding process. Outcome (O): Neural mechanisms and brain regions involved in the mind-decoding process.

With the PICO framework, our research question and hypothesis can be formulated as follows:

**Research Question:** How can fMRI help us understand the neural mechanisms and brain regions involved in mind-decoding tasks in adults?

**Hypothesis:** Employing fMRI in mind-decoding research will reveal specific neural networks and brain regions associated with this cognitive function in adults.

### Search strategy

We used three online databases, including Web of Science (2004-2024), Google Scholar (2000-2024), and PubMed (2009–2024), to identify studies reporting the role of fMRI in mind decoding. We used the following search terms: "fMRI" OR "mind reading/decoding" OR "Neurodevelopmental changes" OR "emotional changes" OR "Social Cognition" OR "Mental State" AND "fMRI". Relevant literature was identified and preserved for subsequent review in order to extract significant information. Only articles published in English were included in the search, and their adherence to the specified inclusion and exclusion criteria was assessed. The online search was conducted by four co-authors between February 1 and March 31, 2024. The search strategy was independently performed by Maher Mohammed Alotaibi and Faisal Saleh Alghamdi. Any disagreements during the screening process were resolved through discussion. If consensus could not be reached,
Sahal Alotaibi and Hamid Osman acted as referees to make the final decision. Afterward, all co-authors collaborated to merge the results and remove any duplicates.

## Selection criteria

The title, abstract, and full text of identified articles were screened independently by all authors to evaluate their eligibility for inclusion in this systematic review. Only studies that met all inclusion criteria were included, and the references cited within those studies were assessed for eligibility. The full texts of the selected studies were downloaded to facilitate thorough decoding and extraction of information, aligning with the primary purpose and objective of this study. Each record was independently screened by Mishaal Abdullah Alshehri, Khaled Majed Bamusa and Ziyad Faiz Almalki, and Sultan Alamri and Ahmad Joman Alghamdi to ensure thorough and unbiased evaluation. Again, any disagreements during the selection process were resolved through discussion. If consensus could not be reached, Sahal Alotaibi and Hamid Osman acted as referees to make the final decision.

## Inclusion and exclusion criteria

To ensure high-quality reporting, we followed the guidelines specified in the Preferred Reporting Items for Systematic Reviews and Meta-analyses (PRISMA) (*Page et al., 2021*). Furthermore, the PRISMA flow chart was utilized to guide the selection process (see Fig. 1).

In this systemic review, because we intend to analyze the role of fMRI in the mind-decoding process, only studies meeting the following criteria were included: (a) the study included only adults; (b) studies published in an open-access or subscription journal; (c) studies published in English language only; (d) studies published on or after 2000; (e) studies that utilize fMRI as a neuroimaging modality; (f) studies that investigate mind decoding or mind-decoding related processes, such as mental state decoding, brain-computer interfaces, or neural correlates of mental processes; (g) studies employing various mind decoding techniques, such as pattern classification, multivariate analysis, or decoding algorithms.

The excluded studies: (a) studies not utilizing fMRI as a neuroimaging modality; (b) studies that do not focus on mind decoding or mind-decoding related processes; (c) studies involving non-human subjects; (d) studies published in languages other than English; (e) review articles, opinion pieces, editorials, and conference abstracts; (f) studies that primarily focus on other neuroimaging techniques or modalities, without substantial emphasis on fMRI and mind decoding.

## Data extraction

The authors conducted an independent screening of the titles and abstracts of the identified articles. Subsequently, full-text articles were thoroughly reviewed to assess their eligibility for inclusion. Data extraction encompassed various aspects of the selected studies, including fundamental characteristics such as authorship, publication date, research type, and methods employed (including fMRI imaging parameters and experimental tasks). Additionally, the essential characteristics of the research subjects, such as age categories (specifically adults), were also extracted.

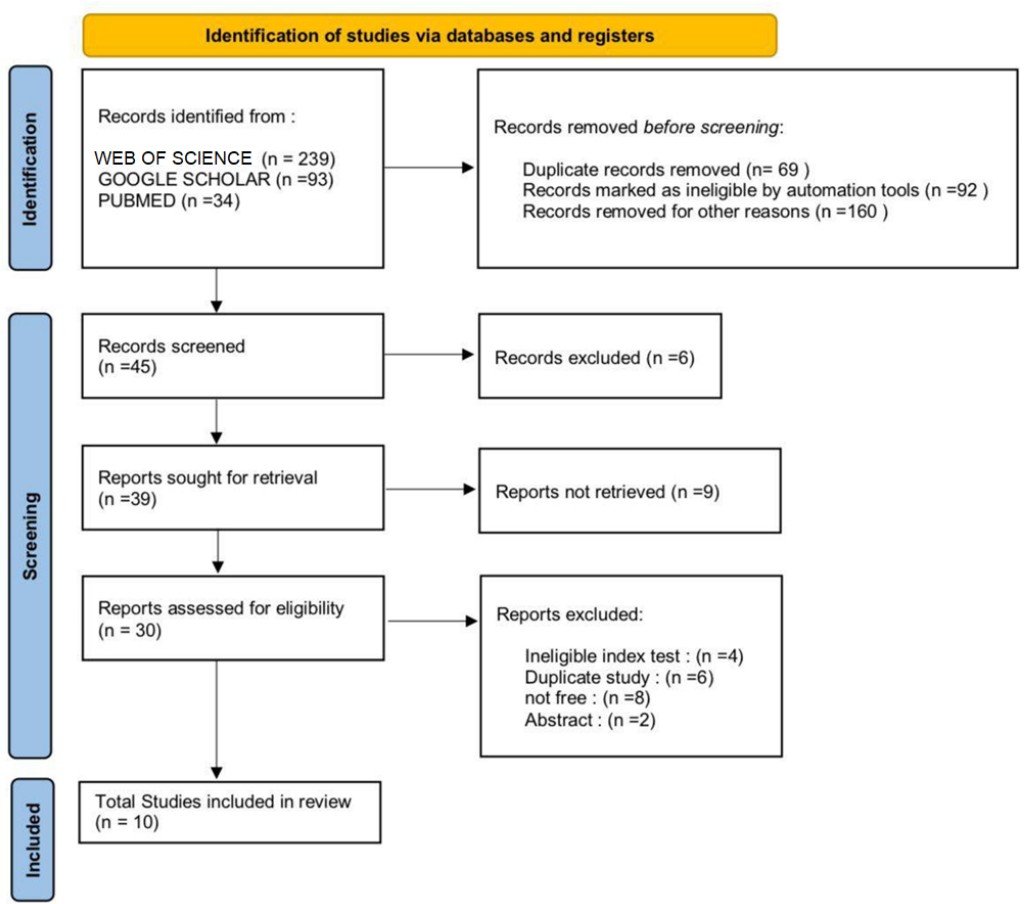

**Figure 1** PRISMA flow chart for the selection process.

## Quality assessment

Each of the authors independently performed a comprehensive quality assessment of the included studies, considering significant factors such as experimental design, statistical methods, reporting quality, methodological rigor, age group categorization, potential sources of bias, and the generalizability of findings.

## RESULTS

### Study selection

A total of 366 articles were initially identified from multiple databases (Web of Science, PubMed, and Google Scholar). After removing duplicates and screening titles and abstracts and paid access, 45 articles were selected for full-text review. Following the full-text review, as a result, a total of 10 studies met the eligibility criteria and were included in this systematic review.

## Characteristics of studies

The included studies were published between 2000 and 2024, covering recent research in the field. All studies focused on adult populations, with participants ranging in age from 18 to 60 years. Various study designs were represented, including cross-sectional, experimental, and longitudinal studies. The studies utilized fMRI as the primary neuroimaging modality to investigate mind decoding or related processes.

## Summary of results related to the role of fMRI in the mind decoding process

The selected studies provided important insights into the topic. (A) One study used the reading & decoding mind in the eyes test (RME) to assess mind decoding among typically developed individuals (*Kawata et al., 2012*). The study calculated differences in Blood-oxygen-level dependent (BOLD) signals between two tasks (A and B) using paired t-tests. The results revealed significant differences in the activation of the right superior occipital gyrus and left parietal lobe, suggesting that these brain regions are specifically involved in mind decoding. These findings were consistent with previous MRI studies using the RME.

(B) Another study focusing on decoding narratives and emotions with fMRI (*Wang et al., 2011*); the main objective was to explore the relationship between anxiety disorders and brain activation during story decoding. The study involved 14 participants with anxiety disorders and 14 control participants. A notable finding was the identification of a significant interaction between the group (anxiety disorder *vs.* control) and emotion in a cluster of brain activation located in the thalamus. Specifically, participants with anxiety disorders exhibited significantly lower levels of thalamic activation compared to the control group. This interaction was detected by comparing brain responses to emotional passages *versus* neutral passages.

(C) This study examined the electrophysiological correlates of decoding the single mind and decoding the interactive mind using Chinese idioms (*Toazza et al., 2022*). The study focused on the activation levels of the medial prefrontal cortex (MPFC), temporal lobe, precuneus, and occipital lobe. The results indicated that these brain regions showed increased activation when stimuli involved two persons compared to stimuli with only one person. These findings aligned with previous fMRI results and event-related potentials (ERPs) studies. Additionally, task-based fMRI studies have reported fMRI signal activities that are linked to specific task; speech sound (*Alotaibi et al., 2023*) and eye movements (*Aloufi, Rowe & Meyer, 2021*) (see Figs. 2 and 3).

(D) In the fMRI study of *Bliksted et al. (2019)*, researchers investigated social cognition in 17 individuals recently diagnosed with first-episode schizophrenia (FES) who had limited or no exposure to antipsychotic medication. These patients were compared to a group of healthy controls matched on a 1:1 basis. The fMRI results revealed that in the control group, intentional movement associated with theory of mind (ToM) activation led to greater activation in specific brain regions, including the temporal gyrus, occipital cortex, and inferior frontal cortex. This activation pattern aligned with previous research findings that observed a response in the posterior superior temporal sulcus

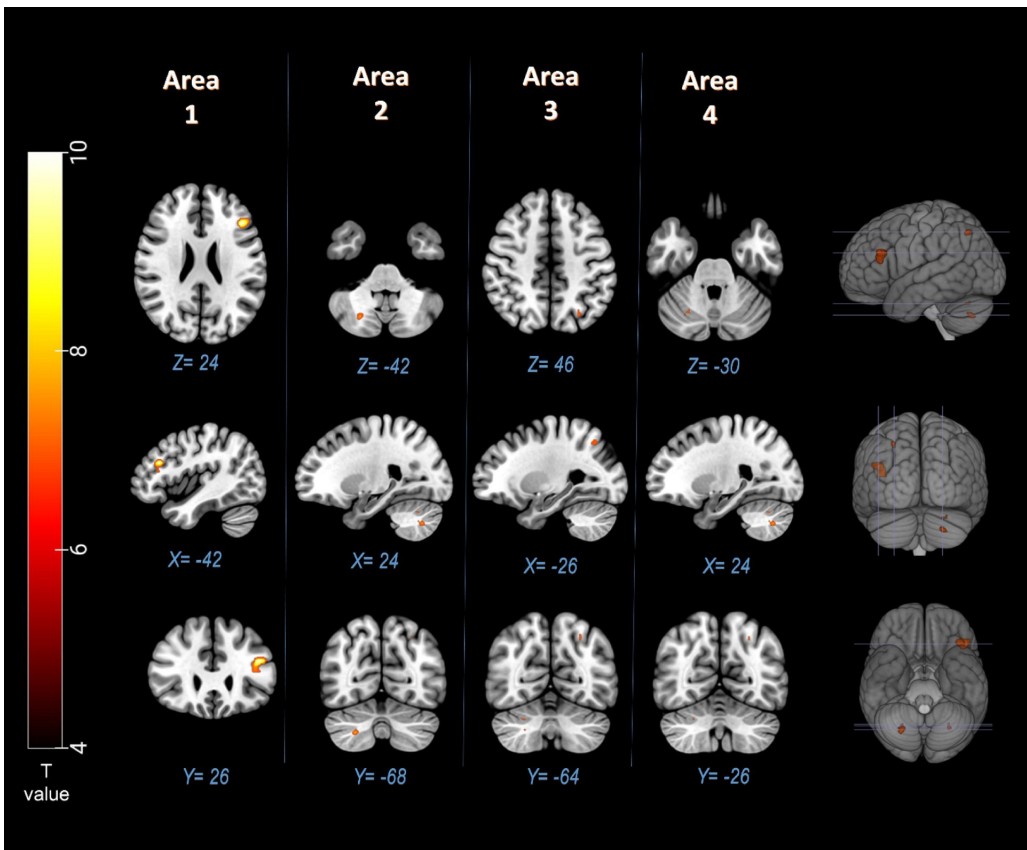

**Figure 2** Significant ($p < 0.05$, FWE) fMRI signal increases in left inferior frontal gyrus, right cerebellum, and left superior parietal lobule (Brodmann area 7) for the speech sound task (*Alotaibi et al., 2023*).

(pSTS) for this particular contrast, validating the expected normal response. Interestingly, the patients with first-episode schizophrenia exhibited a qualitatively similar pattern of activation in these brain regions. However, when comparing the control and patient groups, a distinction was identified in another region, which is further elaborated in the supplementary material (*Bliksted et al., 2019*).

(E) In the study investigating the effects of aging on mind decoding using fMRI (*Castelli et al., 2010*), two groups of healthy adults were included: a young group with an average age of 25.2 years and an old group with an average age of 65.2 years. Participants from both groups underwent fMRI scanning while performing the Decoding the Mind in the Eyes test, which involves attributing mental states to others based solely on their eye expressions. Interestingly, there were no differences in behavioral performance between the young and old groups. Additionally, both groups exhibited activation in the posterior superior temporal sulcus (pSTS) and the temporoparietal junction (TP), indicating that older individuals do not display impairments in the circuits associated with mentalizing processes. (F) In this ongoing study (*Gómez-Carmona et al., 2021*), the primary aim was to examine the brain regions that are activated when individuals process dishes with a
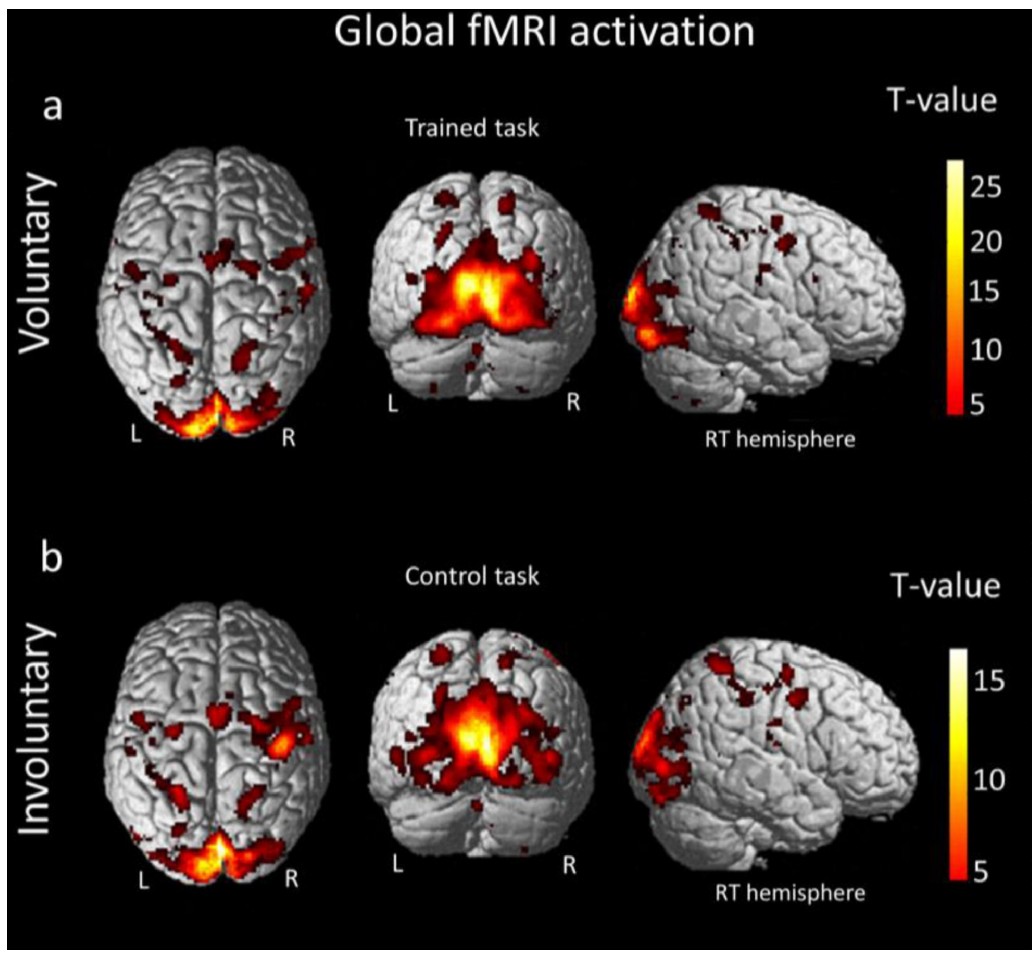

**Figure 3** Global fMRI activation (PFWE < 0.05 cluster level) for the voluntary and involuntary eye movements (*Aloufi, Rowe & Meyer, 2021*).

pleasant design compared to those with an unpleasant design. Additionally, the researchers investigated how previously read rational descriptions compared to emotional descriptions influenced the visualization of the dishes. The study utilized fMRI to analyze brain activity. The findings indicated that participants who visualized pleasant dishes exhibited activation in various domains, including attention, cognition, and reward. Conversely, visualizing unpleasant dishes resulted in stronger activation in regions associated with inhibition, rejection, and ambiguity.

Interestingly, when participants read rational descriptions while visualizing pleasant dishes, brain regions linked to congruence integration were activated. In contrast, participants who visualized emotional descriptions exhibited an increased neuronal response to pleasant dishes in regions associated with memory, emotion, and congruence (*Gómez-Carmona et al., 2021*). (G) In *Underwood et al. (2021)*, the primary aim was to investigate the neural circuits involved in appraising threatening and non-threatening situations in individuals with and without a need for care for psychotic experiences (PEs).

The study included three groups: patients with psychosis spectrum disorder, non-need-for-care participants with PEs, and healthy control participants without PEs. fMRI was used while participants completed the Telepath task, which induced anomalous perceptual experiences. The researchers examined perceived threat evaluations and brain responses during the task, looking for group differences and correlations between brain activation and threat appraisals. The clinical group reported higher subjective threat appraisals compared to both the non-clinical and control groups, while there were no differences between the two non-clinical groups. Additionally, the clinical group exhibited lower BOLD signals in the superior and inferior frontal gyri compared to the non-clinical and control groups. Activation in the precuneus was found to correlate with threat appraisals reported during the task. The findings suggest that intact functioning of fronto-parietal regions may contribute to resilience and the ability to contextualize and evaluate psychotic experiences in individuals with persistent anomalous experiences. (H) In *Pirastru et al. (2023)*, 62 healthy subjects were divided into two groups: group A, consisting of 21 subjects who underwent test-retest reliability assessments on the same day using the same task form, and group B, consisting of 30 subjects who underwent test-retest reproducibility assessments with a 4-month interval using two equivalent-parallel task forms. The results showed that in the voxel-wise general linear model (GLM) analysis, there were no significant differences in reliability and reproducibility for both conditions.

However, when using a region of interest (ROI)-based approach and considering areas with significant main effects of the stimuli, the reliability, as measured by the intraclass correlation coefficient (ICC), was poor ($<0.4$) for the positive condition and ranged from poor to excellent (0.4–0.75) for the negative condition. The ICC-based reproducibility analysis, which compared two different parallel task forms, yielded similar results. (I) In *Reddy, Tsuchiya & Serre (2010)* (RME decoding category), participants completed seven or eight fMRI scanning runs, each lasting approximately 5.6 min. Each run included fixation blocks, visual presentation blocks, and visual imagery blocks. During the visual presentation blocks, participants were shown four categories of objects (food, tools, famous faces, and famous buildings) in separate blocks. Each category was presented twice per run, with four exemplars per category. Each trial within a block consisted of 2 s of visual presentation and 2 s for task response, and the trial order was randomized. The study involved two fMRI experimental conditions: visual presentation (P) and visual imagery (I), both of which were completed by the participants. In the P condition, they viewed exemplars from four stimulus categories (food, tools, famous faces, and famous buildings) in separate blocks. In the I condition; participants heard auditory instructions and were asked to imagine the stimuli. Different exemplars were used in different runs. Prior to the scans, participants familiarized themselves with the stimuli. Activation patterns in the imagery condition were smaller compared to the perception condition (*Reddy, Tsuchiya & Serre, 2010*). (J) In *Tamir et al. (2016)*, a total of 26 participants took part (10 male, 16 female) who were right-handed, native English speakers without any neurological issues. The participants' average age was 21.2 years, ranging from 19 to 26 years. Prior to participation, all individuals provided consent following the guidelines approved by the Committee on the Use of Human Subjects at Harvard University. In this investigation,

we performed initial analyses to pinpoint the brain regions that exhibited responses to two specific characteristics: (i) the vividness of passages describing physical scenes and events, and (ii) whether the passages described a person or a person's mental content. Consistent with previous research, both vivid passages and passages with social content showed increased activation in regions of the default network compared to abstract and non-social passages. Specifically, when comparing social *versus* non-social passages, we found activity in the dorsomedial prefrontal cortex (dmPFC), ventromedial prefrontal cortex (vmPFC), lateral temporal cortex (from the temporal pole to the temporoparietal junction), bilateral hippocampi, and bilateral inferior frontal gyrus (IFG). When comparing vivid *versus* abstract passages, we observed robust activity in the medial temporal lobe (MTL) structures, including the bilateral hippocampus and para-hippocampus, as well as the retro splenial cortex and precuneus (*Tamir et al., 2016*).

Overall, the studies provided valuable insights into the role of fMRI in understanding the mind-decoding process, highlighting specific brain regions and their involvement in various contexts. Individual group activations are further detailed in the Supplementary Information.

## DISCUSSION

The present systematic review has established a robust affirmative correlation between the role of fMRI in understanding mind decoding processes in adults. By analyzing the findings of the included studies, several key themes and implications have emerged.

Firstly, the findings consistently highlight the involvement of specific brain regions and neural networks in mind-decoding tasks. The superior temporal sulcus (STS), medial prefrontal cortex (mPFC), and temporoparietal junction (TPJ) have frequently been implicated in mentalizing processes. These findings align with existing theoretical models that emphasize the importance of these regions in understanding and attributing mental states to others (*Bliksted et al., 2019*; *Castelli et al., 2010*; *Tamir et al., 2016*). However, it is important to note that there is some heterogeneity in the activation patterns observed across studies. This heterogeneity may be attributed to variations in task paradigms and stimuli used. Different mind-decoding tasks, such as perspective-taking or emotion recognition (*Wang et al., 2011*), may engage distinct neural networks, contributing to the observed differences.

Advances in MRI-based brain-decoding research are rapidly expanding our understanding of how neural activity can be translated into meaningful representations of thoughts and imagery. For example, *Tang et al. (2023)* demonstrated the feasibility of decoding continuous language from non-invasive brain recordings, highlighting MRI's potential in capturing complex semantic processing. Similarly, *Koide-Majima, Nishimoto & Majima (2024)* used deep neural network-based Bayesian estimation to reconstruct mental images, underscoring how MRI techniques are advancing toward accurately mapping internal mental states. These studies reflect the critical role of MRI in pushing the boundaries of brain-decoding technology. However, future research should strive for standardized task designs to facilitate better comparability across studies.

Another important consideration is the influence of individual differences on mind-decoding abilities and associated neural activation. Factors such as empathy levels, gender differences, and clinical populations (*e.g.*, autism spectrum disorder) may modulate the observed activation patterns. Future studies should explore these factors systematically and consider their impact on the neural mechanisms underlying mind decoding.

The focus on adult populations in this review is driven by significant differences in neural development and mind-decoding capabilities between adults and children, which have implications for the accuracy and interpretability of fMRI findings. Neural circuits associated with cognitive and social processing, including areas involved in theory of mind and mental state decoding, undergo considerable developmental changes throughout childhood and adolescence. For example, the prefrontal cortex, which plays a key role in complex cognitive functions like perspective-taking and mentalizing, matures later in development, impacting neural responses captured in fMRI studies (*Blakemore, 2012*). These developmental factors suggest that mind-decoding capabilities in children may differ not only in scope but also in neural activation patterns, which could introduce variability when interpreting fMRI data across age groups. Thus, focusing on adults allows for more consistent and interpretable results in the context of mind-decoding research.

Furthermore, the review underscores the potential of fMRI as a tool for developing interventions and therapies for individuals with impaired mind-decoding abilities. For instance, targeted neurofeedback training using real-time fMRI could help enhance mentalizing skills in populations with social cognition deficits. Additionally, understanding the neural underpinnings of mind decoding can inform the development of artificial intelligence systems designed to interpret human emotions and intentions more accurately.

Moreover, the reviewed studies highlight the dynamic nature of mind-decoding processes, suggesting that neural activation patterns may change with experience and training. Longitudinal studies could provide insights into how these neural networks evolve over time and how interventions can lead to sustained improvements in social cognitive functions.

Finally, ethical considerations must be addressed when leveraging fMRI for mind-decoding applications (*Kellmeyer, 2017*). The potential for misuse of this technology in areas such as privacy invasion and surveillance warrant careful regulation and ethical oversight. Researchers and policymakers must collaborate to establish guidelines that ensure the responsible use of fMRI in both clinical and non-clinical settings.

## CONCLUSION

This systematic review finds consistent involvement of specific brain regions (dmPFC, TPJ, STS, mPFC) during mind decoding tasks, indicating a neural basis for social cognition. Variations in activation patterns and methodological differences are identified, emphasizing the need for standardized protocols. The review suggests using identified neural correlates to inform interventions for individuals with social communication difficulties and explores the potential clinical applications of fMRI in enhancing mind-decoding abilities.

## LIMITATIONS

- The review may have only captured published studies, potentially missing unpublished data with negative or inconclusive results.
- Focusing only on English-language studies might exclude valuable research from other languages.
- Limiting the search to 2000-2024 might miss more recent advancements in the field.

### Limitations of mind-decoding techniques

- Decoding thoughts and intentions from brain activity using fMRI is still under development and may not always be accurate.
- The effectiveness of mind-decoding techniques using fMRI might be influenced by the specific tasks used in the studies. Techniques might not generalize to real-world scenarios.
- As fMRI technology advances, ethical concerns arise regarding privacy and potential misuse of mind-decoding capabilities.

## ACKNOWLEDGEMENTS

During the preparation of this work the author(s) used some online service/tools such as Grammarly, ChatGPT 4.0 in order to improve language and readability. After using this tool/service, we reviewed and edited the content as needed and take full responsibility for the content of the publication.

### Funding

The authors received no funding for this work.

### Competing Interests

The authors declare there are no competing interests.

### Author Contributions

- Sahal Alotaibi conceived and designed the experiments, authored or reviewed drafts of the article, and approved the final draft.
- Maher Mohammed Alotaibi performed the experiments, analyzed the data, prepared figures and/or tables, and approved the final draft.
- Faisal Saleh Alghamdi performed the experiments, authored or reviewed drafts of the article, and approved the final draft.
- Mishaal Abdullah Alshehri performed the experiments, prepared figures and/or tables, and approved the final draft.
- Khaled Majed Bamusa analyzed the data, authored or reviewed drafts of the article, and approved the final draft.
- Ziyad Faiz Almalki conceived and designed the experiments, analyzed the data, prepared figures and/or tables, authored or reviewed drafts of the article, and approved the final draft.

- Sultan Alamri analyzed the data, prepared figures and/or tables, and approved the final draft.
- Ahmad Joman Alghamdi performed the experiments, analyzed the data, prepared figures and/or tables, and approved the final draft.
- Mohammed Alhazmi analyzed the data, authored or reviewed drafts of the article, and approved the final draft.
- Hamid Osman conceived and designed the experiments, performed the experiments, analyzed the data, authored or reviewed drafts of the article, and approved the final draft.
- Mayeen U Khandaker conceived and designed the experiments, authored or reviewed drafts of the article, and approved the final draft.

## Data Availability

This is a systematic review/meta-analysis.

## Supplemental Information

Supplemental information for this article can be found online at http://dx.doi.org/10.7717/peerj.18795#supplemental-information.

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
