# Peer review of "The role of fMRI in the mind decoding process in adults: a systematic review"

_PeerJ, doi:10.7717/peerj.18795_

## Round 0.1 · original submission · Major Revisions

Thank you for submitting your manuscript. We have carefully reviewed your work and appreciate the insights it contributes to the field. The reviewers have identified several concerns that must be addressed for the manuscript to reach its full potential. We kindly request that you respond to each of these points in detail. In particular, please revise the term "mind-reading" to "mind-decoding" throughout, as this terminology more accurately reflects the nature of your work.

·

Basic reporting

The authors have conducted a comprehensive review of fMRI studies related to “mind-reading” technology. Their systematic manner and detailed summaries would provide helpful information to researchers in the field of both machine learning and neural engineering. I have several concerns and questions on the current manuscript as follows:

1) Why did they present EEG results in Figures 2 and 3 while the main topic of this review manuscript is fMRI. It would be helpful to clarify the rationale for emphasizing these results and how they contribute to the overall understanding of “The role of fMRI in the mind reading process”.

2) Recent studies listed below have demonstrated the feasibility of reconstructing (decoding) imagined images and speeches from fMRI signals. These advancements are highly relevant to the topic of 'mind-reading' and, even if they don't strictly meet the authors' selection criteria, noting them would be helpful to readers interested in fMRI-based mind-reading technology.

Semantic reconstruction of continuous language from non-invasive brain recordings. Jerry Tang, Amanda LeBel, Shailee Jain, Alexander G. Huth. Nature Neuroscience. 2023. https://doi.org/10.1038/s41593-023-01304-9
Mental image reconstruction from human brain activity: Neural decoding of mental imagery via deep neural network-based Bayesian estimation. Naoko Koide-Majima, Shinji Nishimoto, Kei Majima. Neural Networks. 2024. https://doi.org/10.1016/j.neunet.2023.11.024

3) While this review manuscript focuses on “adults”, some people would think that the difference between adults and children is not so large in the context of mind-reading technology. Why did the authors specifically focus on adults? What kind of potential difference does exist between them? It would be beneficial to explain it in the introduction or discussion section.

Experimental design

NA. This is a review paper.

Validity of the findings

NA. This is a review paper.

Reviewer 2 ·

Basic reporting

Statements in line 103-118 which leads to questions whether this review is rigorous. When referring to the references 7-11, words including "likely" and "might" raises doubts whether the articles cited within were understood or even read by the authors. Please read these references to confirm whether or not these statements are true.

Figures are included without appropriate use permission statements.

Experimental design

no comment

Validity of the findings

no comment

·

Basic reporting

Overall the English language is clear. However authors need to provide clear and concise definition of what they accept as "mind-reading". The currently employed formulations are somewhat ambiguous and may mislead the reader into para-psychological direction of interpretation. Given that in essence this review is dedicated to social cognition and mentalization I would recommend the use of more neutral notions, such as "mind-decoding".
The referenced lietarature is up-to-date and provides relatiovely sound context. The article structure, and figures are appropriate.

Experimental design

Research question is not well defined, as it has been outlined earlier. It has to be explained which specific psychological functions and dysfunctions are identified as knowledge gap in the exisiting body of literature. The investigations apprears to be rigorous, following the established PRISMA guidelines and PICO framework. Methods are described in sufficient details to allow replciation.

Validity of the findings

The impact of the paper is conditional, dependent on more precise formulation of the goals and rationale.
All underlying data have been provided and seem to be robust.
Conclusions are well stated and supported with results.

Additional comments

None

---

## Round 0.2 · Minor Revisions

The reviewers acknowledged that the manuscript has improved after revision. However, there are still some minor issues that need to be addressed. Please review the comments from Reviewer 1.

·

Basic reporting

The manuscript has been improved, and the authors have addressed many of the concerns raised in the previous reviews. However, I think that the authors use the coined phrase "fMRI mind-reading (decoding)" with two different meanings without clarification:

1) using fMRI to directly decode cognitive states or intentions from neural activity patterns; and
2) investigating how individuals infer the mental states of others based on their behavior or neural activity.

These are two different topics and, individual studies related to either of them were explained without clear distinction, which may lead to confusion. For example, studies (h) and (i) are respectively corresponding to these two, but they were almost seamlessly explained in the manuscript. To improve the clarity of the manuscript, I recommend:

- Providing a clear definition of "fMRI mind-reading/decoding" in the introduction section. I think that it is OK if the definition here is a mixture of the two.
- Clearly delineating the two types of studies throughout the manuscript, perhaps using different headings or subheadings.


I also think that the manuscript could benefit from a final proofread by a native English speaker. Several expressions seem awkward or unclear. For example, on lines 359-360, the authors wrote "The results revealed that Participants underwent two fMRI experimental conditions:", which is not a result but rather a description of the experimental procedure.

Experimental design

NA. This is a review paper.

Validity of the findings

NA. This is a review paper.

·

Basic reporting

The basic reporting has been improved according to the comments of peer review report.

Experimental design

The research question has been better defined, following the peer review report.

Validity of the findings

No further comments.

---

## Round 0.3 · Minor Revisions

The authors did not address the second recommendation of Reviewer 1: "Clearly delineating the two types of studies throughout the manuscript, perhaps using different headings or subheadings." Implementing this change is important to enhance the clarity of the manuscript for readers.

---

## Round 0.4 · accepted · Accept

The authors addressed all the reviewers concerns. The manuscript is now ready for publication.